



# Deglacial sea-level history of the East Siberian Sea Margin

Thomas M. Cronin[1], Matt O'Regan[2], Christof Pearce[2], Laura Gemery[1], Michael Toomey[1], Igor Semiletov[3,4], Martin Jakobsson[2]

[1] US Geological Survey MS926A, Reston, Virginia, 20192, USA
[2] Department of Geological Sciences and Bolin Centre for Climate Research, Stockholm University, Stockholm, 10691, Sweden
[3] Pacific Oceanological Institute, Russian Academy Sciences, Vladivostok, Russia
[4] Tomsk National Research Polytechnic University, Tomsk, Russia

*Correspondence to*: Thomas M. Cronin (tcronin@usgs.gov)

**Abstract.** Deglacial (12.8-10.7 ka) sea-level history on the East Siberian continental shelf/upper continental slope was reconstructed using new geophysical records and sediment cores taken during Leg 2 of the 2014 SWERUS-C3 expedition. The focus of this study is two cores from Herald Canyon, piston core SWERUS-L2-4-PC1 (4-PC) and multicore SWERUS-L2-4-MC1 (4-MC1) and a gravity core from an East Siberian Sea Transect, SWERUS-L2-20-GC1 (20-GC). Cores 4-PC1 and 20-GC were taken at 120 m and 115 m modern water depth, respectively, only a few meters above the global last glacial maximum (LGM, ~ 24 kiloannum (ka)) minimum sea level of ~ 125-130 meters below sea level (mbsl). Using calibrated radiocarbon ages mainly on molluscs for chronology and the ecology of benthic foraminifera and ostracode species to estimate paleo-depths, the data reveal dominance of river-proximal species during the early part of the Younger Dryas event (YD, Greenland Stadial GS-1) followed by a rise in river-intermediate species in the late Younger Dryas or the early Holocene (Preboreal) period. A rapid relative sea-level rise beginning roughly 11.4 to 10.8 ka (~ 400 cm core depth) during is indicated by a sharp faunal change and unconformity or condensed zone of sedimentation. Regional sea level at this time was about 108 mbsl at the 4-PC1 site and 102 mbsl at 20-GC. Regional sea-level during the YD was about 40 to 50 meters lower than those predicted by geophysical models corrected for glacio-isostatic adjustment. This discrepancy could be explained by delayed isostatic adjustment caused by a greater volume and/or geographical extent of glacial-age land ice and/or ice shelves in the western Arctic Ocean and adjacent Siberian land areas.

**Keywords.** Arctic Ocean, deglacial sea level, Younger Dryas, glacio-isostatic adjustment

## 1 Introduction

Rates and patterns of global sea-level rise (SLR) following the last glacial maximum (LGM) are known from radiometric ages on coral reefs from Barbados, Tahiti, New Guinea, and the Indian Ocean and sediment records from the Sunda Shelf and elsewhere. These records provide estimated global and regional rates of SLR when combined with LGM and deglacial ice-sheet history and geophysical models of regionally varying glacio-isostatic adjustment (GIA) to changing land ice mass. For example, Lambeck et al. (2014) estimates mean global rates during the main deglaciation phase 16.5 to 8.2 kiloannum (ka) at 12 mm yr$^{-1}$ with more rapid SLR rates (~40 mm yr$^{-1}$) during Meltwater Pulse 1A ~14.5-14.0 ka and slower rates during the Younger Dryas (YD) from 12.5-11.5 ka. Importantly for our discussion, Lambeck et al. (2014) do not find evidence for rapid SLR during Meltwater Pulse 1B ~ 11.3 ka. The ICE-6G sea-level model of Peltier et al. (2015, see also Argus et al. 2014) also provides spatially varying rates of vertical land motion focused on regions of major ice sheets of North America, Fennoscandinavia and Antarctica and regions peripheral to ice sheets. ICE-6G provides a general model against which new regional sea-level curves may be compared.



Our initial working hypothesis is that the regional sea-level record from the East Siberian margin would exhibit behavior like that expected in "peripheral bulge" areas located 1000-3000 km from the centers of LGM ice sheets. These areas would include areas like the Mid-Atlantic region of eastern North America (Cronin et al. 2007) and Europe (Lambeck et al. 2011, Steffen and Wu 2011) that were isostatically uplifted during peak glaciation and subsequently subsided in a collapsing forebulge. However, the western Arctic Ocean and the adjacent Siberian margin are relatively poorly known in terms of both regional sealevel and glacial history (Klemann et al. 2015). It is especially important to investigate the SLR history of this region in light of recent submarine geophysical and sediment core evidence for extensive ice-shelf and iceberg scouring during glacial periods Marine Isotope Stages (MIS) 6, 4 and 2. Submarine evidence comes from across much of the Arctic Ocean including the Hovgaard Ridge–Arctic Ocean (Arndt et al. 2014), the Beaufort Sea (Engels et al. 2008), Chukchi Sea (Polyak et al. 2007, Dove et al. 2014) and East Siberian Sea (Niessen et al. 2013) margins, the Lomonosov Ridge, the Arlis Plateau and the slope off the Herald Canyon, East Siberian Sea (Jakobsson et al., 2010, 2016). These new glacial discoveries suggest the need for a reevaluation of basic assumptions that underpin sea-level reconstructions based on geophysical models, marine planktic and benthic oxygen isotopes, and coral reefs.

In addition, although studies of deglacial sediments using micropaleontological proxies are common throughout the Arctic and subarctic regions, most studied cores are from water depths deeper than 200 m (Ishman et al. 1996, Scott et al. 2009, Taldenkova et al. 2013, Carbonara et al. 2016, Hald 1999, Osterman et al. 2002. Jennings et al. 2014, Wollenburg et al. 2007, Knudsen et al. 2004). The proxies provide excellent paleoceanographic records of water-mass history but not direct evidence for the elevation and location of paleoshorelines. One exception is the study by Rasmussen and Thomsen (2014, see also Hald et al. 2004) of Storfjorden, Svalbard that demonstrates the sensitivity of foraminiferal species to rapid deglacial and Holocene climate changes in a previously glaciated area. Unlike the Svalbard region, the East Siberian margin did not host a large LGM ice sheet (Svendsen et al., 2004; Jakobsson et al., 2014), but instead was located some distance from the Fennoscandinavian Ice Sheet and North American ice sheets (Peltier et al. 2014, Klemann et al. 2015). Thus, at least for much of the Younger Dryas and post-Younger Dryas intervals (12.8-10.7 ka), results presented here provide insight not only into Arctic sea level and glacial history and GIA models but also extra-Arctic sea level records used to estimate global sea level patterns.

## 2 Material and Methods

This study is a result of the 2014 SWERUS-C3 expedition (Swedish–Russian–US Arctic Ocean Investigation of climate–cryosphere–carbon interactions) to the Chukchi Sea, Herald Canyon, East Siberian Sea Margin, and southern and central Lomonosov Ridge. The results are derived from ship-based and post-cruise, shore-based analyses of the stratigraphy, physical properties, chronology and micropaleontology of sediment cores taken along two transects in the Chukchi Sea and East Siberian Sea across the outer continental shelf and upper continental slope.

### 2.1 Stratigraphy and physical properties

Cores SWERUS-L2-4-PC1 and 4 MC (4-PC1, 4-MC, 72° 50.3447' N, 175° 43.6383' W) were part of a continental shelf and upper continental slope transect from the Herald Canyon region of the Chukchi Sea designed to recover Holocene and pre-Holocene deglacial sediments (Table 1, Figure 1). The sediment bulk density and magnetic susceptibility (MS) (Figure 2) for the Herald Canyon transect clearly show a downcore increase in both MS and density in cores 4-PC1, beginning about 380 cm core depth, and in 5-GC at ~ 80 cm (Figure 2). Core SWERUS-L2-2-PC1 (2-PC1, 77° 21.5370'N 163° 02.0226' E) from 57 meters below sea level (mbsl) on the shelf, recovered Holocene sediments with a maximum age of 4.2 ka (Pearce et al., this volume). The focus of the current study is the interval in core 4-PC1 between 350 and 602 cm core depth and from 90-120 cm in core 5-GC, which both record the late deglacial regional SLR during and following the Younger Dryas. Multicore 4-MC was used to identify late Holocene microfaunal assemblages, which were distinct from those deposited during the late postglacial interval.



The East Siberian Sea transect included four cores from 115 m to 964 m below sea level (Figure 3). Our focus was on samples from the core catcher in core 20-GC, about 56-81 cm below the seafloor, which recorded a transitional unit with increasing MS and density downcore, similar to the trend seen in core 4-PC1. The chirp sonar profile crossing the coring site of 4-PC1 reveals acoustic characteristics suggesting that denser, coarser-grained post-glacial sediments may have

prevented further penetration and core recovery (Jakobsson et al. this volume). This is also likely for the 5-GC and 20-GC sites. As discussed below, radiocarbon dating confirms that transitional units at both the Herald Canyon and East Siberian Sea margin represent nearly coeval late deglacial sediments deposited during sea-level transgression. The other three cores in the East Siberian Sea transect were taken from water depths that are too deep to observe a sedimentological or micropaleontological signal of regional sea level.

## 2.2 Micropaleontology

Benthic foraminifera (Edwards and Wright 2015) and ostracodes (Cronin 2015) are among the more useful groups for reconstructing paleo-sea level records due to the restricted environmental preferences of many species. The benthic foraminifera and ostracodes from cores 4-PC1, 4-MC, 5-GC and 20-GC were studied in order to reconstruct

paleoenvironmental conditions and specifically to estimate paleo-depth ranges from key indicator species (Supplementary Material). The working halves of the sediment cores were sampled shipboard using 20 cm$^3$ plastic scoops. Bulk sediment samples were sealed in labeled plastic bags and stored refrigerated. Samples were processed shipboard to assess microfossil preservation and to target productive intervals, which were later sampled at Stockholm University in 2015. All samples were washed through a 63-μm sieve using a light-diffusive pressure stream to disaggregate the sediment and collect the sand size

fraction. The remaining coarse fraction was rinsed and decanted out of the sieve using distilled water from a squirt bottle on to labeled filter paper, then dried in the oven at ≤50°C for a minimum of 5 hours. Sediment was stored in 15 ml snap-cap glass vials and detailed micropaleontological studies of benthic foraminifera and ostracodes were conducted at the US Geological Survey (USGS) in Reston Virginia.

The following literature sources on Arctic Ocean foraminiferal taxonomy and ecology were used: Wollenburg and Mackensen (1998), Polyak et al. (2002), Scott et al. (2008), and McDougall (1994). The most useful publication on shallow-water Arctic foraminiferal species' ecology is Polyak et al. (2002), which we used as a primary source of key indicator species for inner shelf, river-influenced environments. The river-proximal assemblage of Polyak et al. (2002) from the southern Kara Sea was critical for indicating modern analog assemblages for nearshore, brackish-water environments influenced by freshwater river influx from the Ob and Yenisey Rivers. Korsun and Hald (1998), Wollenburg and Kuhnt

(2000), and Osterman et al. (1999) also were consulted and provided complementary ecological data from other regions of the Arctic Ocean.

The primary sources of ostracode taxonomy and ecology were Stepanova (2006), Yasuhara et al. (2014), and Gemery et al.
(2015) and references therein. The study by Gemery et al. (2015) gives modern depth range and ecological data for 1340 modern surface samples from the Arctic and subarctic seas. Simple statistical analyses were used to obtain the most reasonable paleo-depth estimates from fossil biofacies on the basis of modern ecology and modern depth ranges of key species.


## 2.3 Radiocarbon chronology

From core SWERUS-L2 samples containing the planktonic foraminifera *Neogloboquadrina pachyderma*, mixed benthic foraminifera or mollusk shells (identified by A. Gukov) were picked shipboard for accelerator mass spectrometry (AMS)
radiocarbon measurements upon return. Further radiocarbon sampling was conducted at Stockholm University and the USGS, Reston, Virginia and ages were obtained from the National Ocean Sciences Accelerator Mass Spectrometry (NOSAMS) facility at Woods Hole Oceanographic Institution, Woods Hole, Massachusetts as well as from the Radiocarbon Dating Laboratory at Lund University and Beta Analytic.




The chronology of core SWERUS-L2-4-PC1 is based on 10 AMS radiocarbon ages, including one outlier (Fig. 4) presented and discussed in Jakobsson et al (this volume). On the basis of a major change in the physical properties and geochemical composition of the core (Figure 2), a key lithologic transition from ~413-400 cm depth was identified that includes a hiatus or condensed section. All ages are calibrated with the Marine13 calibration curve (Reimer et al., 2013) using the age depth modelling software Oxcal 4.2 (Bronk Ramsey, 2008, 2009). Two different values are used for the local marine radiocarbon reservoir correction. In the lower part of the core, up to approximately 400 cm, analyses of sediment chemical and physical properties suggest that there is no connection to the Pacific Ocean and thus no inflow of relatively old Pacific waters (Jakobsson et al., this volume). For this lower section, a $\Delta R = 50 \pm 100$ years was applied on the basis of present values in the Laptev Sea (Bauch et al., 2001), the closest site with modern information on the reservoir age from a shallow, coastal Arctic shelf setting with no Pacific influence (Reimer and Reimer, 2001). In the upper 400 cm of the core, which represent the late Holocene, a larger reservoir was expected due to the Pacific influence and a $\Delta R = 300 \pm 200$ years was applied to the radiocarbon ages in this section. This value is lower than the 477 years derived for neighboring core 2-PC1 (Pearce et al., 2016) because of the significantly greater water depth of the site. Below 100 meters water depth, Atlantic-sourced waters are present in the Herald Canyon (Linders et al., 2015) and these waters have a lower radiocarbon reservoir age. Six radiocarbon ages were also obtained from the corecatcher of core 20-GC and one each from 23-CG and 24-GC and are given in Table 2 (see Fig. 4).

## 3. Results

### 3.1 Herald Canyon

Age model

The age of the base of the core at 609 cm composite depth is estimated to be approximately 13.5 ka, although this is based on extrapolation beyond the lowermost date at 499 cm. This implies that the entire Younger Dryas stadial (~12.9 – 11.7 ka; Steffensen et al., 2008) is captured in core 4-PC1, between approximately 460 – 560 cm depth. The lower section of Core 4-PC1 is characterized by a nearly uniform high sediment accumulation rate of about 85 cm/ka, based on dated levels between 417 and 499 cm, yielding ages around 11100 – 12100 ka (Figure 4). The age of the midpoint of the transition between the lower and upper lithological units at 407 cm is estimated to (1-sigma age range) 10787 – 11209 cal yrs BP. Jakobsson et al (this volume) refer to the upper and lower units of Core 4-PC1 as Units A and B, respectively. The transition between the two units was defined based on measured $\delta^{13}C_{org}$ in the sediments (Jakobsson et al., this volume). The age range was determined using a $\Delta R = 50$ and upcore extrapolation of the youngest age at 417 cm depth. The upper Unit A of Core 4-PC1 extends to present day. The oldest dated level in Unit A is at 399 cm where a 1-sigma age range of 8332 – 8805 cal yrs BP was acquired from dating unidentified organic material. An age model was not developed for individual samples in core 20-GC however the estimated age of the key micropaleontological data are discussed below.

Micropaleontology

Figure 5 shows the river-proximal and river-intermediate foraminiferal species from core 4-PC1 in the upper and lower panels, respectively. The most noteworthy feature of the pre- to early Younger Dryas interval is the dominance of river-proximal species (*Elphidium bartletti, Haynesina orbiculare*) below 510 cm. The subsequent decrease in this assemblage is coincident with an increase to 40-50% in river intermediate species such as *Cassidulina reniforme*, *Pyrgo williamsoni*, and *Quinqueloculina seminulum* (Fig. 5; Supplementary Appendix). This transition is somewhat abrupt, beginning ~ 520 cm core depth, possibly marking the onset of the Younger Dryas. River-intermediate species remained common in the interval 520 – 400 cm of core 4-PC1, with the age of the transition from 413 to 400 cm core depth estimated to be about 11 ka (Jakobsson et al., this issue). Sedimentation increased at this site during the late Holocene ~ 3.4 ka (Fig. 4). It is noteworthy that, like the ostracode assemblages discussed below, Holocene shelf foraminiferal faunas are dominated by fully marine, mid-to-outer shelf species and contrast strongly with late deglacial assemblages dominated by species signifying riverine influence.





Analyses of benthic ostracode from core 4-PC1 yielded the similar paleoenvironmental results; the modern geographic distributions and depth ranges of key species are shown in Figure 6 (Supplementary Appendix). From the base of the core at 609 cm to ~ 509 cm there are rare ostracodes, mainly *Paracyprideis pseudopunctilata* and *Sarsicytheridea punctillata*, both

shallow nearshore species. From 504 cm to 427 cm there is an abundant and diverse assemblage including *Cytheromorpha macchesneyi*, *Rabilimis* sp., *Cytheropteron* spp., *Acanthocythereis dunelmensis* (mean modern depth of 16 m) and *Semicytherura complanata*. *C. macchesneyi* and *P. pseuopunctillata* (mean modern depths of 18 m) represent ~ 23 and 29 % of 4-PC1 assemblages from 500 to 460 cm core depth, the interval containing with river intermediate foraminiferal assemblages. From 367 to 352 cm there were rare specimens of *A. dunelmensis*, *Kotoracythere arctoborealis*, and

*Cytheropteron* spp. similar to Holocene shelf assemblages found in multicore 4-MC and in 2-PC1 from 57 mwd (Pearce et al. this volume). Thus, there is a faunal change from shallow-water, brackish assemblages to mid-shelf assemblages near the aforementioned unconformity spanning the late deglacial-Holocene boundary.

Core 5-GC core catcher samples from ~ 100-125 cm core depth contain a calcareous benthic assemblage with common

specimens of *E. bartletti* and *H. orbiculare*, both river proximal species found in the lowermost zone of 4-PC1. In contrast, samples from 15 to 75 cm core depth contain typical Holocene shelf benthic foraminifers (mainly *Elphidium excavatum clavata*) and ostracodes (*A. dunelmensis*, *Elofsonella concinna*, *Normanocythere leioderma*, *Cytheropteron* spp.).

3.2 East Siberian Slope

Age model

In core 20-GC (115 mwd), 6 AMS radiocarbon dates between 56 and 81 cm indicate ages between ~13 and 11 ka (Figure 4B, Table 2), and the average of the median ages of six radiocarbon ages is 12.0 ka. This sequence contains several reversals

in radiocarbon ages, and it is more likely for old material to be reworked and redeposited, compared to the contamination of younger material into underlying sediments. The age of this unit therefore is probably around 11.0 ka (the youngest series of dates in this core) with significant input of older, reworked material.

Micropaleontology


Microfaunas from this interval suggest deposition in a shallow nearshore environment. For example, the foraminifers *Elphidium bartletti*, *Haynesina orbiculare* and *Elphidium incertum* are river-proximal species in the Kara Sea (Polyak et al. 2002). The ostracodes *Heterocypridies sorbyana*, *Rabilimis* sp., and *Sarsicytheridea punctillata* all tolerate reduced salinity near river mouths and estuaries. On the basis of modern species distributions (Gemery et al. 2015), the mean depth range for

*H. sorbyana* is 14 m (n=90) (Figure 6). In the Laptev and Kara Seas *Rabilimis* is found in modern samples of similar depths (Stepanova 2006). These nearshore ostracode species are either absent or occur in low numbers at deeper sites in the Laptev and Kara Seas.

**4. Discussion**

4.1 Comparison with other Arctic deglacial microfaunal records

The results show a faunal similarity of diagnostic nearshore assemblages in East Siberian Sea cores taken on the outer shelf/upper slope (core 4-PC1 ~520-600 cm, core 5-GC ~100-125 cm and core 20-GC ~56-81 cm) supports the evidence from the physical properties (higher sand content, high magnetic susceptibility and bulk density; Figures 2, 3) for shallow

marine environments about 13.5 to 12 ka. The lithologic transition between Unit B1 and B2 at 520-510 cm coincides with the faunal shift in foraminiferal assemblages (Fig. 5). The core 4-PC1 river-intermediate assemblage centered about 12 ka suggests an early to mid Younger Dryas transgression of this region. Above this assemblage, there is an unconformity or condensed interval, presumably after the final phase of post-glacial SLR had breached the Bering Strait and fully submerged the continental shelf (Jakobsson et al., this volume). Sediment accumulation rates increased again during the late Holocene.




There are additional benthic foraminiferal records from the Eurasian shelf/slope indicating oceanographic changes during the Younger Dryas-early Holocene intervals. Figure 7 compares the benthic foraminiferal data from core 4-PC1 with the foraminiferal data from core PS51/154 (270 mwd) from the Laptev Sea (Taldenkova et al. 2013). We used the assemblage scheme of river-proximal, river intermediate and river distal taxa defined by Polyak et al. (2002). Gray shading marks key climatic intervals identified by Taldenkova et al. (2013) and suggested by the informal foraminiferal zones in core 4-PC1. River proximal species, which dominate from 530 to 600 cm core depth in core 4-PC1 on the East Siberian Sea margin, are, in contrast, generally rare (usually < 10%) in core PS51/154. This change in abundance is expected due to the greater water depth (270 m) and distance from shore for PS51/154 compared to Core 4-PC1 (120 m). Overlying the river proximal zone in Core 4-PC1 is an interval with greatly increased proportions (40-50%) of river intermediate species suggesting deeper water due to regional sea-level transgression. This assemblage, dated at 12 to 10.7 ka, was deposited during the late-Younger Dryas and transition into the Holocene (Preboreal) period.

Coeval oscillations in foraminiferal assemblages in core PS51/154 coincide with those in 4-PC1, however those in core PS51/154 signify water mass changes on the upper slope rather than changes in salinity due to a greater rate of SLR and greater distance to fresh water river influx. Late Holocene foraminiferal assemblages in core 4-PC1 (not shown) are typical of many Arctic shelf environments and are dominated by *E. excavatum clavata*.

### 4.2 Regional sea-level datums: Deglacial sea level in the western Arctic

Relative sea level datums were calculated for intervals in cores 4-PC1 and 20-GC dated at approximately 13.5-12.5 ka as follows. The depth of core interval containing the [14]C-dated, nearshore ostracode and river-proximal foraminiferal assemblages (600-520 cm core depth for core 4-PC1) was added to modern water depth of 120 m at the core site, and then the paleo-water depth (17 m on the basis of preferred modern depths for *C. macchesneyi*, *P. pseudopunctillata*, *Rabilimis*) was subtracted. This gives a paleo-shoreline datum near 13.5 -12.5 ka at about 108 mbsl at the Herald Canyon site. Doing a similar calculation for core 20-GC, the river-proximal assemblage (mean core depth 73 cm) plus water depth at core site (115 m), minus paleodepth (~14 m based on *H. sorbyana*), yields a paleo-shoreline of 102 m. These estimates of 108 m and 102 m should be viewed as approximate, although they are based on a large ecological literature given above. The possibility exists the datum for core 20-GC is several centuries younger if one omits the two radiocarbon ages from possibly reworked shells. The post-Younger Dryas (post-11.7 ka) rate of SLR cannot be quantified but very likely it was faster than the rate during the Younger Dryas on the basis of coral reef records.

It is useful to compare relative sea level for the late deglacial period with estimated sea level for this region from the ICE-6G (VM5a) geophysical model of Peltier et al (2014, Argus et al. 2014). ICE-6G (VM5a) is the latest version of geophysical models of Earth surface boundary conditions from the LGM and most recent deglaciation in response to the change in mass distribution as ice sheets melted. The newest version has additional refinements applying Global Positioning System (GPS) measurements of vertical crustal movements, which are verified by recent Gravity Recovery and Climate Experiment (GRACE) satellite data. Expected paleo-water depth histories near our core sites were determined in two steps: (1) the modern (t = 0 ka) depth below sea-level was calculated as the sum of the shipboard core collection water depth and sample sediment depth for each SL datum. No additional correction for sediment compaction was made. (2) Then, we subtracted the modeled ICE-6G topography difference from present (Topo_Diff) at time 't' from the value calculated in step 1. This approach was used in place of the absolute model topography due to its coarse discretization in the model (1 °latitude x 1 °longitude).

For the East Siberian Sea, the ICE 6G model predicts a paleo-water depth of 64 mbsl around ~11.7 ka. Paleo-depth estimates reported above on the basis of micro-faunal assemblages diverge sharply from modelled, deglacial sea level estimates (Figure 8). For cores 4-PC1 and 20-GC, the modelled depths for ~11.7 ka are respectively 47 m and 42 m deeper than the estimates based on micro-faunal depth zonation zonation and chronologic constraints described above. While a



fraction of this offset might be explained by hydro-isostasy (e.g. Klemann et al., 2015), this discrepancy might also be reconciled by (1) the presence of East Siberian ice cover, perhaps on the continental shelf  (2) a forebulge along the ice periphery and (3) a subsequent collapse following deglaciation.  At present little evidence for LGM ice has been found in the East Siberian or Chukchi Seas and previous suggestions for circum-Arctic glaciation (Grosswald and Hughes, 2002) conflict

with apparent 'ice-free' LGM conditions on Wrangel Island (71 °N, 179 °W) and elsewhere in Eastern Siberia (e.g. Gualtieri et al., 2003 and references therein). However, given that surveys only recently identified widespread scouring by a large ~1 km-thick Arctic ice shelf during recent glacial periods (Polyak et al., 2001; Niessen et al. 2013, Jakobsson et al., 2016), the results presented here give cause for continued exploration in search of LGM glacial landforms submerged along Arctic continental margins.

Two major conclusions can be drawn from the new findings from Leg 2 of the SWERUS-C3 expedition: (1) late deglacial regional sea level along the East Siberian Sea margin during the Younger Dryas was roughly 42-47 m lower than levels expected from geophysical models of glacio-isostatic response to the last deglaciation; and (2) there appears to be evidence from both the Herald Canyon and East Siberian Sea margin sites of a relatively rapid rise in sea level following the Younger

Dryas. Although it is difficult to estimate the rate of SLR, the age seems to correspond to Meltwater Pulse 1B and would thus be considered evidence for global sea level rise.

## 6. Acknowledgements

We are grateful to the captain and crew of the *Oden* for assistance in carrying out the SWERUS C3 Leg 2 expedition. Special thanks go to Jan Backman for shipboard direction, Carina Johansson, Natalia Macho Barrientos, Pedro Pietro, for

shipboard laboratory work, and to A. Gukov for mollusc identifications. M. Robinson and C. Swezey provided useful reviews of the manuscript.  Funded by US Geological Survey Climate and Land Use R&D Program and the Knut and Alice Wallenberg Foundation (KAW) for the SWERU-C3 expedition. Additional support from the Swedish Research Council (Jakobsson; 2012-1680, O'Regan; 2012-3091) and the Danish Council for Independent Research (Pearce: grant no. DFF-4002-00098_FNU).

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

**Figure Captions**

**Fig 1. Map**
Map showing the approximate cruise tract of the SWERUS Leg 2 2014 expedition and the locations of cores SWERUS L2-4-PC1 and SWERUS L2-20-GC1 used in this study. Details on the cores can be found in Table 1.

**Fig 2. Herald Canyon Transect, Cores 2-PC1, 4-PC1, 5-GC1**
Transect of Herald Canyon cores SWERUS-L2-2-PC1, 4-PC1 and 5-GC showing core photographs, lithologic units A and
B, radiocarbon age control (red diamonds), magnetic susceptibility and bulk density. Further details on the age model of core 2-PC1 can be found in Pearce et al. (this volume) and additional details on the lithology of 2-PC and 4-PC in Jakobsson et al. (this volume)

**Fig 3. East Siberian Sea Transect, Cores 20-CG, 22-PC, 23-GC, 24-GC**
Transect of the East Siberian Sea cores SWERUS-L2-20-GC, 22-PC, 23-GC, and 24-GC crossing the shallow shelf (20-GC) and the slope of the East Siberian Sea (22-PC, 23-GC, 24-GC) showing core photos, radiocarbon date locations, magnetic susceptibility and bulk density. Radiocarbon dates indicate that the deglacial, dense, high susceptibility, dark grey sediments recovered in the lower half of 20-GC are not correlative to a similar lithology at the base of 22-PC, 23-GC and 24-GC, which pre-date the LGM. They are potentially correlative to a thin higher susceptibility and dense grey sediment layer found in the
upper 50 cm of the slope cores.

**Fig 4a. Radiocarbon corrected age model** for core SWERUS-L2-4-PC1 showing unconformity about 400 cm core depth.
4b. **Radiocarbon corrected age model** for core SWERUS-L2-20-GC1.

**Fig 5. Core 4-PC1 foraminiferal species**
The distribution of benthic foraminifers in the SWERUS-L2-4-PC1 core showing species' percent abundance and depth of the samples in the core. Upper panel shows river proximal species, lower panel river intermediate species. Ecological classification for species groups is from Polyak et al. (2002). In the river-proximal species group, *Elphidium bartletti* is common. In the river-intermediate species group, *Cassidulina reniforme* is dominant. Shading estimates the boundary





between the late deglacial (Younger Dryas, Preboreal) and the Holocene intervals. See also Figure 7. Stratigraphic units A and B are based on multiple criteria and discussed in Jakobsson et al. this volume).

**Fig 6. Modern depth ranges and modern geographic distribution for ostracode species used for paleodepth estimation**
**for core 4-PC1.**
The ostracodes a) *Cytheromorpha macchesneyi,* b) *Paracyprideis pseudopunctillata* and c) *Heterocyprideis sorbyana* (from 4-PC) are plotted showing percent abundance of each species at modern depths based on the 1200-sample modern Arctic Ostracode Database (AOD) (Gemery et al. 2015). Only modern AOD samples with >50 total ostracode specimens were used. *Rabilimis* (not shown) is also a marginal marine indicator species found in core 20-GC (see Gemery et al. this volume).

**Fig 7. Core 4-PC1 deglacial foraminiferal assemblages compared to those from the Laptev Sea**
Comparison of benthic foraminifers in cores SWERUS-L2-4-PC1 and PS51/154 from the Laptev Sea (Taldenkova et al. (2013) using the assemblage scheme of river proximal, river intermediate and river distal taxa defined by Polyak et al. (2002). Shading marks key climatic intervals identified by Taldenkova et al. (2013) and informal foraminiferal zones in core
4-PC. River proximal species dominate from 530 to 600 cm core depth in 4-PC1 on the East Siberian Sea margin. In contrast, river proximal species are generally rare (usually < 10%) in PS51/154 due to the greater water depth (270 m) and distance from shore for the core site compared to SWERUS 4-PC1 (120 m). Overlying the river proximal zone in 4-PC1 is an interval with greatly increased proportions (40-50%) of river intermediate species suggesting deeper water due to regional sea-level transgression. This zone is dated at 12 to 10.7 ka, mainly the post-Younger Dryas and transition into the Holocene
(Preboreal) period. Oscillations in foraminiferal assemblages in core PS51/154 coincide with those in SWERUS 4-PC1, however those in PS51/154 signify water mass changes on the upper slope rather than salinity changes due to changes in the rate of sea level rise and proximity to fresh water river influx. Late Holocene foraminiferal assemblages at 4-PC (not shown) are dominated by *E. excavatum clavata*.

**Fig 8. Comparison of deglacial relative sea-level positions from SWERUS cores and modeled RSL.**
a) Map showing modern, Younger Dryas and LGM shorelines based on the ICE 6G model of Peltier et al., (2015). LGM (24 ka) ice sheet extent and thickness are designated by blue shading. A green 'X' and orange dot mark location of core sites 20-GC1 and 4-PC1, respectively. b) Orange dot and green 'X' show paleo Younger Dryas (about 11.7 ka) water depth estimates based on micro-faunal assemblages. Expected paleo-water depth histories for 20-GC1 and 4-PC1 are given by the
blue solid and red dotted line, respectively.





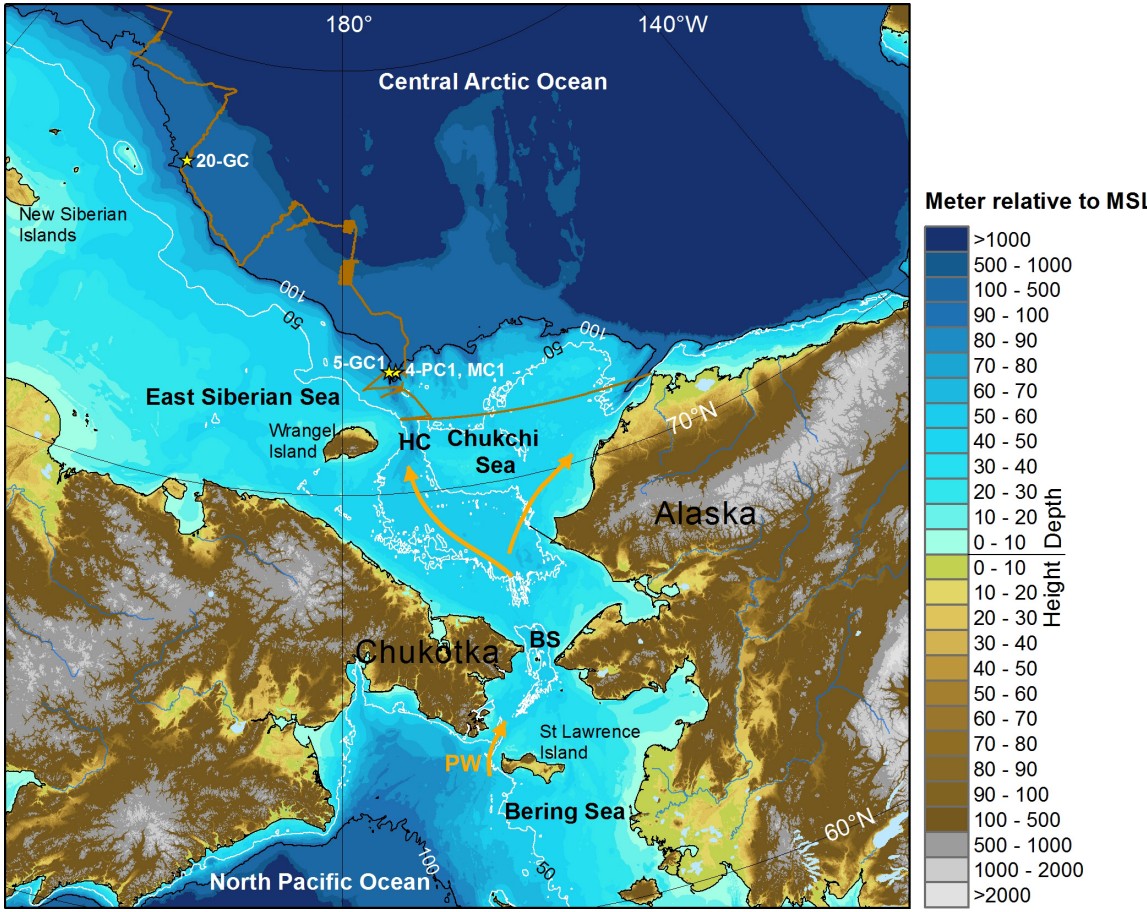

Figure 1



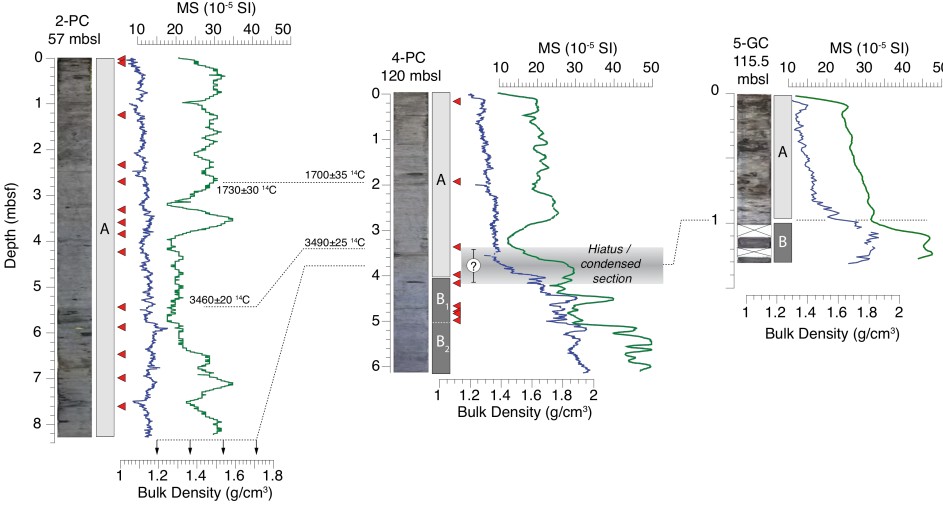

Figure 2




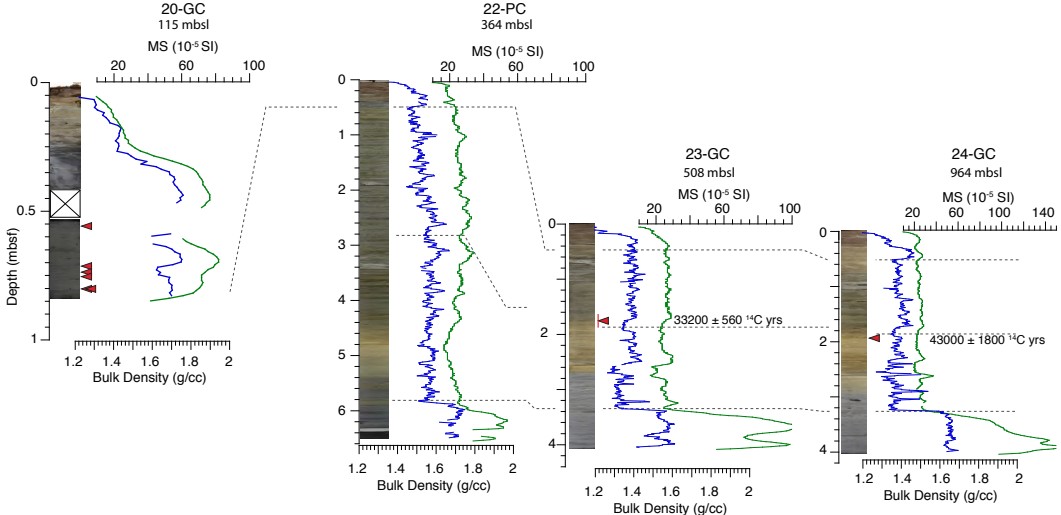

Figure 3





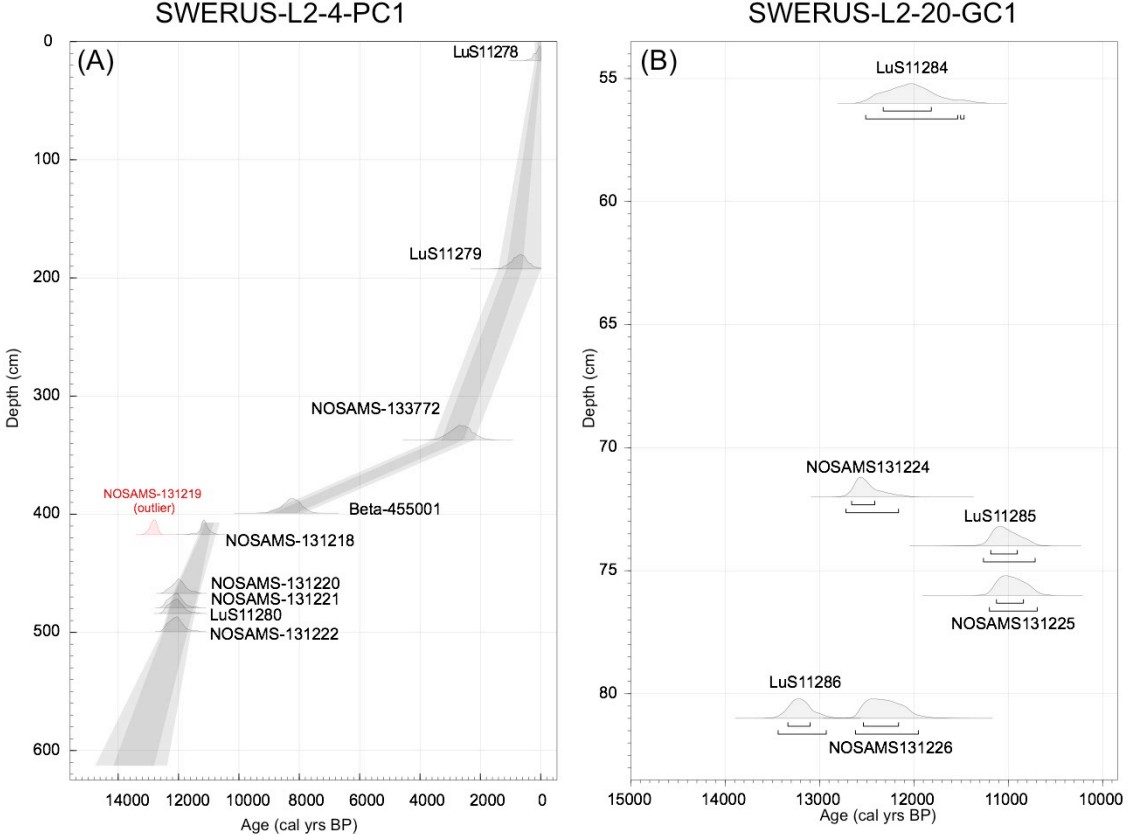

Figure 4



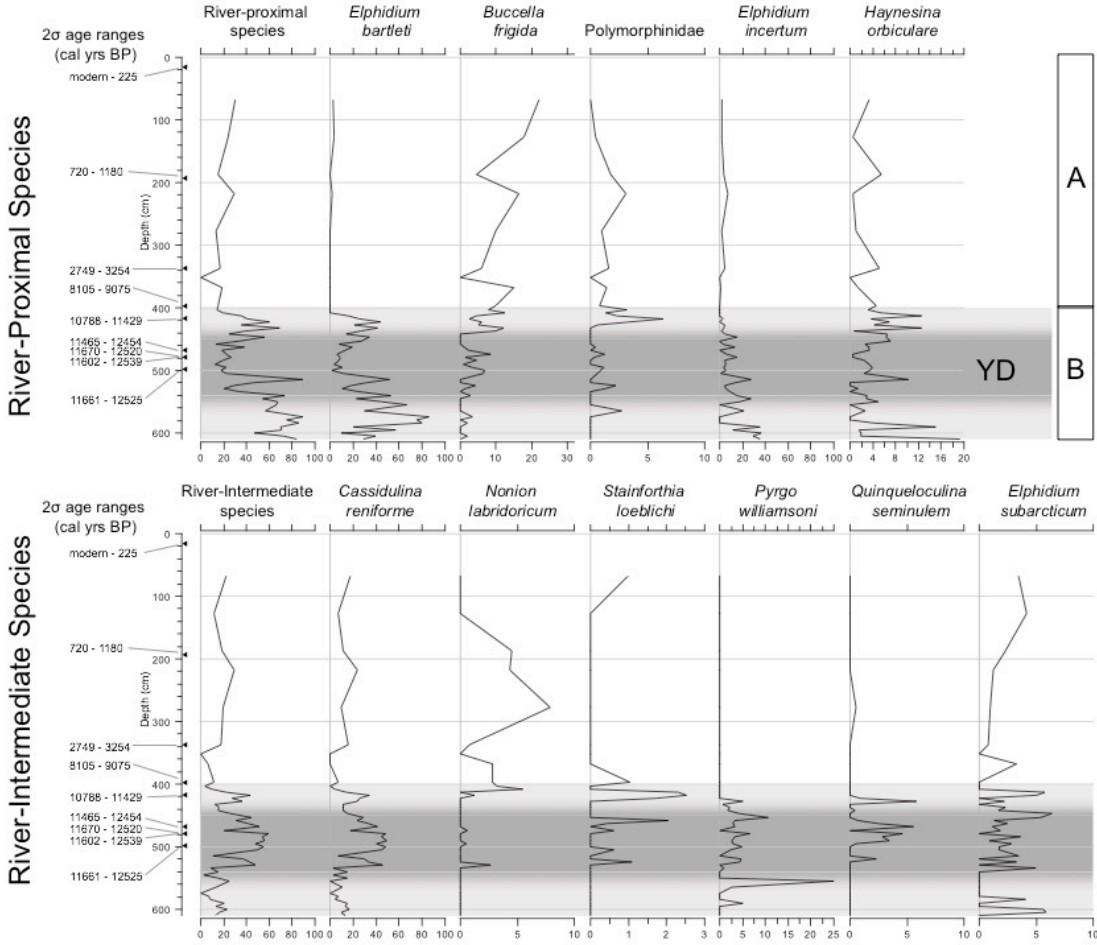

Figure 5





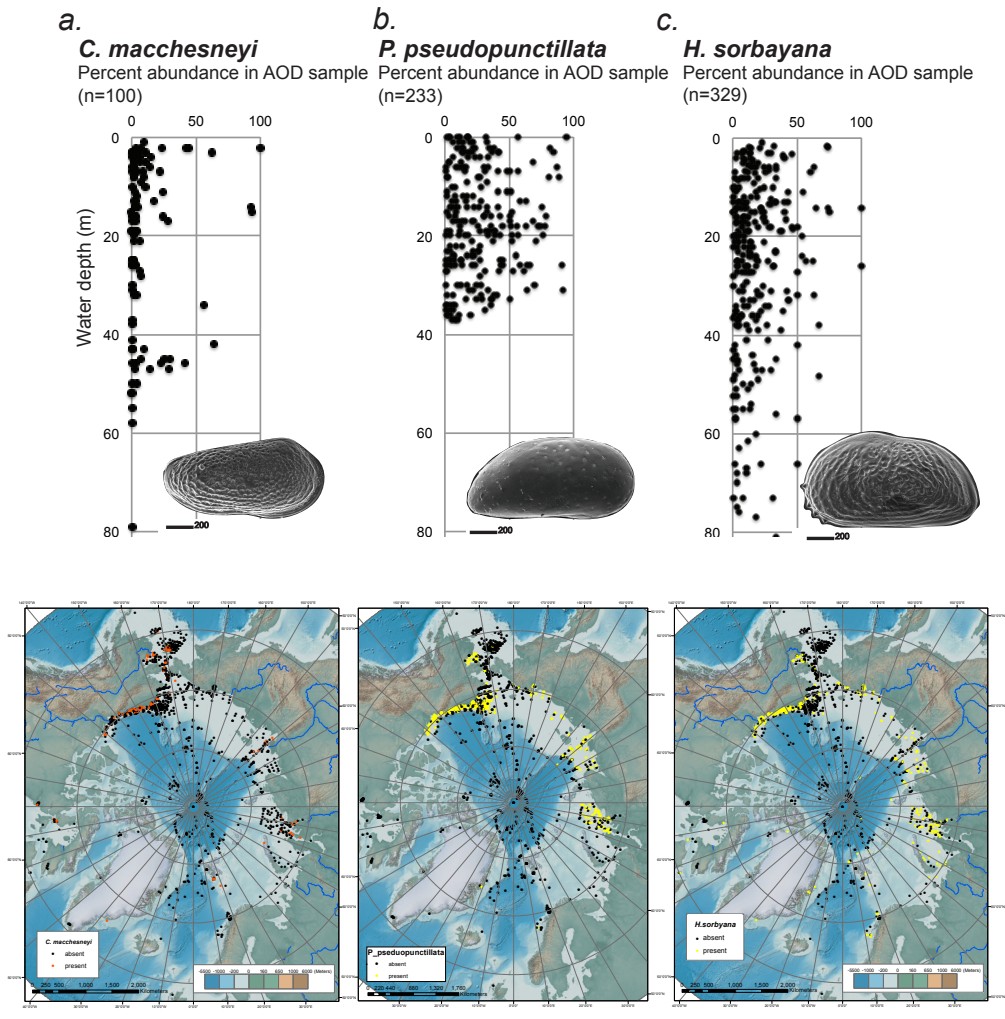

Figure 6



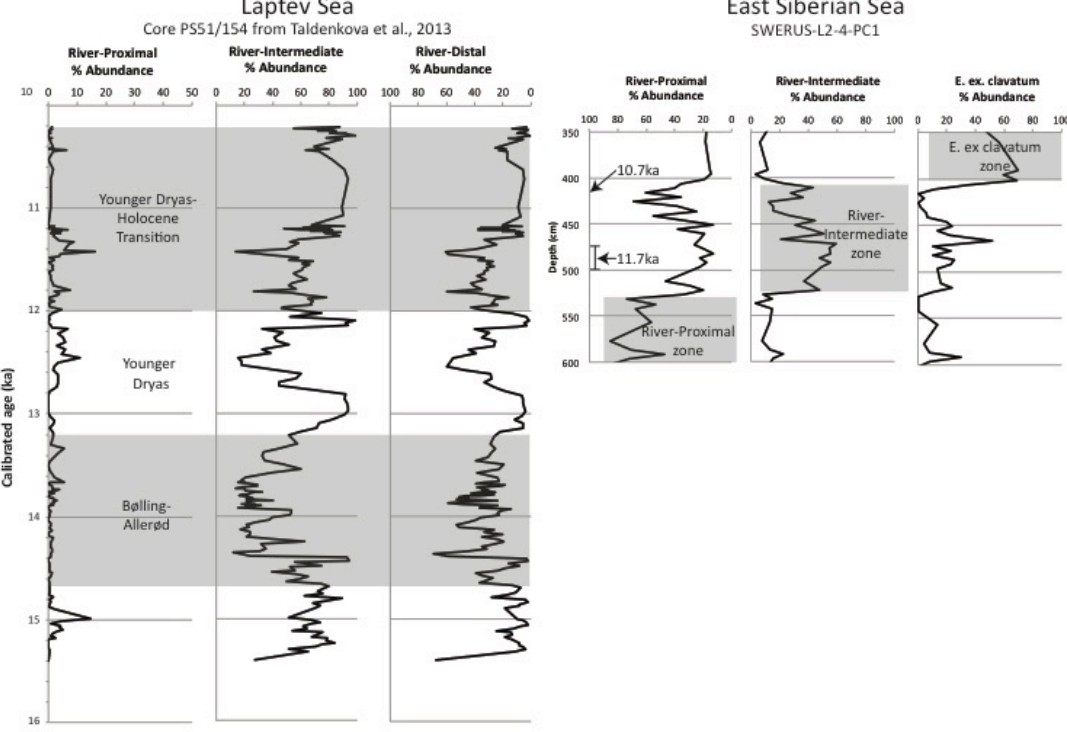

Figure 7




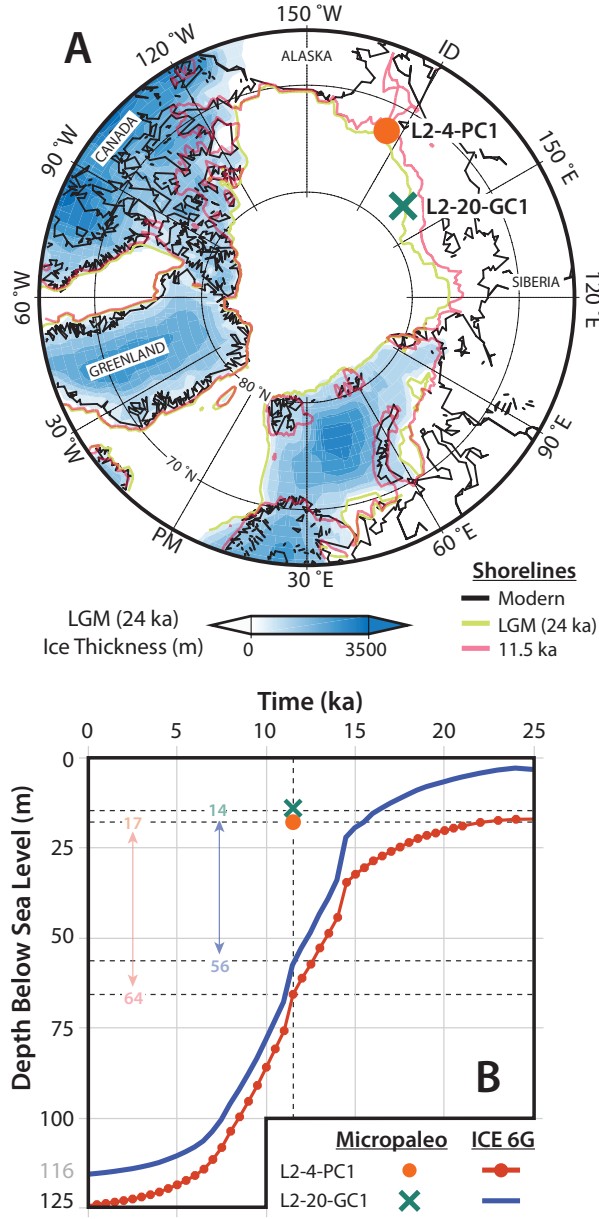



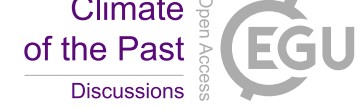

Figure 8

Tables

Table 1. SWERUS C3 Leg 2 Core Data

| Core ID | Lat (N)_aft dec | Long_aft deck | LatDec_from tim | LonDec_from tin | WD (m) | Length (m) | Recovery (%) |
|---|---|---|---|---|---|---|---|
| SWERUS-L2-2-PC1 | 72° 30.001' | 175° 19.170' W | 72.516580 | -175.319605 | 57 | 8.24 | 83.6 |
| SWERUS-L2-4-PC1 | 72° 50.3447' | 175° 43.6383' W | 72.8387 | -175.727122 | 119.7 | 6.13 | 68 |
| SWERUS-L2-4-MC1 | 72.8608333333 | -175.7108 | 72.86083333 | -175.7108333 | 123.6 | 0.1 | NA |
| SWERUS-L2-5-GC1 | 72° 52.1769' | 176° 12.4630' W | 72.869910 | -176.207350 | 115.5 | 1.43 | 48 |
| SWERUS-L2-20-GC | 77° 21.5370' | 163° 02.0226' E | 77.359052 | 163.033304 | 115 | 0.83 | 14 |
| SWERUS-L2-22-PC | 78° 13.3728' | 164° 27.7194' E | 78.222926 | 164.461842 | 364 | 6.45 | 72 |
| SWERUS-L2-23-GC | 78° 39.6556' | 165° 00.9492' E | 78.660932 | 165.015603 | 508 | 4.06 | 68 |
| SWERUS-L2-24-GC | 78° 47.81544' | 165° 21.9861' E | 78.796922 | 165.366530 | 964 | 4.05 | 68 |

Table 2. Radiocarbon ages calibrated from SWERUS C3 Leg 2, -4-PC, 20-GC, 23-GC, 24-GC

| Core | Lab ID | mid core depth (cm) | material dated | C14 age (yrs BP) | Error | ΔR | Calibrated, unmodelled 2 sigma from (cal yrs to (cal yrs BP) | | mean | error | median | Modelled age mean | error |
|---|---|---|---|---|---|---|---|---|---|---|---|---|---|
| 4-PC1,1, 15-17 cm | LuS11278 | 16 | Mollusc: *Nuculan* | 445 | 35 | 300 ± 200 | 225 | ... | 65 | 54 | 51 | 105 | 79 |
| 4-PC1,2, 141-142 cm | LuS11279 | 192.5 | Mollusc: *Yoldia a* | 1700 | 35 | 300 ± 200 | 1180 | 720 | 954 | 117 | 952 | 1236 | 166 |
| 4PC1,3, 135-137 cm | NOSAMS13377 | 337 | Mollusc | 3490 | 25 | 300 ± 200 | 3254 | 2749 | 3003 | 131 | 2998 | 2835 | 243 |
| 4-PC1, 4, 65-67 cm | NOSAMS13121 | 417 | Mollusc | 10200 | 30 | 50 ± 100 | 11429 | 10788 | 11139 | 146 | 11143 | 11112 | 147 |
| 4-PC1,4, 65-67 cm | NOSAMS13121 | 417 | Mollusc | 11400 | 35 | 50 ± 100 | 13070 | 12635 | 12834 | 112 | 12826 | Outlier | - |
| 4-PC1, 4, 115-117 cm | NOSAMS13122 | 467 | Mollusc | 10700 | 30 | 50 ± 100 | 12454 | 11465 | 11987 | 237 | 11994 | 11870 | 268 |
| 4-PC1, 5, 5-7 cm | NOSAMS13122 | 479 | Mollusc | 10750 | 30 | 50 ± 100 | 12520 | 11670 | 12095 | 220 | 12101 | 11964 | 264 |
| 4-PC1,5, 10-12 cm | LuS11280 | 484 | Mollusc: *Yoldia a* | 10745 | 55 | 50 ± 100 | 12539 | 11602 | 12078 | 238 | 12088 | 11993 | 264 |
| 4-PC1,5, 25.27 cm | NOSAMS13122 | 499 | Mollusc | 10750 | 35 | 50 ± 100 | 12525 | 11661 | 12094 | 222 | 12100 | 12079 | 267 |
| 20-GC1,CC, 2-4 cm | LuS11284 | 56 | Mixed benthic for | 10725 | 65 | 50 ± 100 | 12511 | 11468 | 12034 | 254 | 12044 | | |
| 20-GC1,CC, 18-20 cm | NOSAMS13122 | 72 | Mollusc | 11050 | 30 | 50 ± 100 | 12720 | 12163 | 12490 | 142 | 12521 | | |
| 20-GC1, CC, 20-22 cm | LuS11285 | 74 | Mollusc: *Macoma* | 10110 | 55 | 50 ± 100 | 11263 | 10715 | 11020 | 145 | 11034 | | |
| 20-GC1,CC, 22-24 cm | NOSAMS13122 | 76 | Mollusc | 10050 | 40 | 50 ± 100 | 11200 | 10693 | 10958 | 136 | 10968 | | |
| 20-GC1,CC, 27-29 cm | LuS11286 | 81 | Mollusc: *Macoma* | 11785 | 65 | 50 ± 100 | 13439 | 12929 | 13200 | 126 | 13209 | | |
| 20-GC1,CC, 27-29 cm | NOSAMS13122 | 81 | Mollusc | 10900 | 60 | 50 ± 100 | 12619 | 11953 | 12302 | 182 | 12318 | | |
| 23-GC1, 2, 62-79 cm | Lu131228 | 169-186 | Mollusc | 33200 | 560 | | | | | | | | |
| 23-GC1, 2, 87-89 cm | Lu131229 | 192 | Foraminifera, plan | 43000 | 1800 | | | | | | | | |