# Peer review of "Deglacial sea-level history of the East Siberian and Chukchi Sea Margins"

_Climate of the Past, 2017_

## Short Comment (SC1) · 17 Apr 2017

This is an interesting study that attempts to use ostracods of known environmental affiliation to estimate paleosea-level along the continental shelf of the Arctic Ocean. However, as with so many studies of sediment cores from the Arctic Ocean, this paper is bedeviled by dating problems, making the conclusions untenable.

The entire interpretation of the lower record rests on undated sediments that are assumed to be of "Younger Dryas" age (∼12.9-11.7ka BP) by extrapolation from a date at 417cm (11,112) and 4 samples below it (which are essentially of the same age over 32cm [467-499cm: range in mean age is 11,870-12,079, and range in median age is 11,987-12,094). In this way, the authors conclude that the base of the core, at 609cm, is "approximately 13.5ka". Based on the interpreted age of 13.5ka at the base of the

record, the authors then conclude that regional sea-level was 40-50m lower than geophysical models predict, but that discrepancy is entirely based on the assumption that the age extrapolation to the base of the core is correct. A simpler explanation is that it is not correct, and that the basal sediments are older, a possibility that is not considered. The authors mention (without discussion) that "Unit B' has 2 facies (B1 and B2). The first time this is mentioned is in Section 4.1. This "lithologic transition" does not enter into their assumption of a linear sedimentation rate below the lowest date (which appears to be in unit B1) but may explain why the sediments towards the base of the core are older than assumed.

It also appears that the 417cm sample is immediately below a hiatus of unknown duration in the core. A hiatus in the record seems highly likely. From ∼11.5-11.0ka B.P., (MWP-1B) sea level rose by 16m (>3m/century). During this time, water depths at the core site would have been quite shallow, and it is hard to imagine that sediment deposition in this dynamic environment was not completely disturbed. It also seems unlikely that the ∼8500 B.P date (on unidentified organics) is correct as this would imply a dramatic reduction in sediment deposition from ∼2835 to ∼8500 BP., followed by a sharp increase.

The discussion of core 20-GC [Section 3.2] is bizarre as there is no consistency in the dates on that record, and the authors simply decide to ignore older ages as being reworked, concluding that the entire record is "probably around 11ka". Similar logic is not applied to old dates on samples dated in cores 23-GC and 24-GC—these are accepted as correct.

There is also a puzzling use of reservoir corrections—300 years for the upper section, but only 50 years for the lower section. In a recent paper–on which the first author here was a co-author (Poirier et al, 2012, Marine Micropaleontology) a reservoir age of 1,000 years was used for samples >10,000 years, as Hanslik et al., 2010 (QSR) also did. One might expect that restricted circulation in the Canada Basin, prior to the opening of Bering Strait, would result in "old water" in this area, requiring a bigger

reservoir correction. That would shift the age of the radiocarbon dates in Unit B1 towards younger calibrated ages.

Other points: Inconsistencies in core IDs (in text, Figure). Also appear to be errors in Lab ID of C-14 dates in Table [some numbers are duplicated and Beta-455001 in Fig 4 not listed]. This often makes it a challenge to follow the arguments in the paper.

The conclusions reached in the paper are unconvincing.

---

## Referee Comment (RC1) · Anonymous Referee #1 · 26 Apr 2017

The paper "Deglacial sea-level history of the East Siberian Sea Margin" has the aim to reconstruct and estimate, from sediment cores, the paleo-depth in the eastern margin using benthic foraminifera and ostracode species.

Thought the results are discussed properly and they could be relevant for a large audience, I am not convinced by the dating of the cores. Without a confident age calibration all the discussion given in the manuscript are of secondary importance. The results obtained by the authors can be much improved if the age scale and the confidence associate with the dating of the core will be improved.

The chronology of the lower section of core PC1 is based only on 5 points (6 with the outlier) where four of them suggest a similar age, though they cover 33 cm of the core. Considering this uncertainties and lack of age calibration points in the bottom

part of the core I would be careful in assume an almost linear relationship depth vs age to date the bottom part, especially for the time periods investigate. The periods between 10 ky and 15 ky BP in the Arctic, have been characterize by rapid climate changes brought to different climate condition. In this period we can distinguish three different climate phases, the Younger Dryas, the Bølling-Allerød interstadial and the early Holocene. These periods could have been characterized by different deposition and sedimentation rate making the assumption of a linear sedimentation rate below the lowest date not convincing.

The core 20-GC1 shows a similar age between 55 to 90 cm depths. Could the continuity of these sediments be disturbed and reshuffle by the Fennoscandinavia ice sheet movement?

Without additional information improving the core chronology, especially for the bottom part of the PC1 core, the discussion present by the authors are not convincing.

---

## Referee Comment (RC2) · E. Taldenkova (Referee) · 29 Apr 2017

This interesting paper addresses an important question about the last postglacial sea-level variability in the eastern Eurasian Arctic seas in relation to the recently obtained new evidence on glacial ice influence in this part of the Arctic Ocean and its margins. It is clearly written and well-structured, the discussion and conclusions are conceivable.

However, I have several concerns about the interpretation of the data presented.

1. My first concern is the dating of sediment sequences.

- The authors refer to Bauch et al., 2001 when they explain the application of $\Delta R=50\pm100$ years for the oldest section in core 4-PC1 from the Herald Canyon that was not affected by Pacific waters. However, in the paper of Bauch et al., 2001 the

average ∆R for the Laptev Sea based on the measurements of live molluscs collected prior to 1950 and stored in the Zoological museum was estimated as 370 yrs (see their Table 1).

- When estimating the age of the section in core 20-GC from the East Siberian Sea margin I would rather rely on the dating obtained on mixed benthic foraminifers from 56 cm. The whole sediment section is bioturbated, and infaunal molluscs like Macoma could have burrowed into older sediments, like in the case with the datings at 72, 74 and 76 cm. I would suggest for the age model to take the dating on forams at 56 cm (10725 14C), the dating at 72 cm (11050 14C), and then the old dating at 81 cm (11785 14C). Thus, the time span for sediment accumulation would be rather of 12-13 cal.ka, i.e. the YD.

2. My major concern is the interpretation of the species composition of benthic foraminifers and ostracods in terms of reconstructing past water depths.

- First of all, from the point of view of statistically correct interpretation, I wouldn't calculate percentages of species in the samples that contain less than 100 foram tests, but rather present their abundance in the form of tests/g dry weight. In fact, almost all samples from the sediment section of core 4-PC1 below 504 cm contain less than 100 tests (see Supplementary material). The same is true for several samples from the upper sediment units. Ostracods are usually rather rare in sediments from Arctic shelf seas and slope. This is also the case with the samples from the current study. Most of them from both localities contain less than 10 valves. There is a slightly more abundant interval in core 4-PC1 between 504 and 427 for which the authors calculated relative abundances of species, but actually only 4 samples from this interval contain more than 20 valves, whereas 4 samples are barren of ostracods.

- All samples from core 20-GC that contain river-proximal foraminifers and euryhaline ostracods Paracyprideis pseudopuctillata and Heterocyprideis sorbyana do also contain abundant river-intermediate species and some relatively deep-water species like

Islandiella (Cassidulina) teretis among forams which is an indicator of transformed Atlantic waters in the Arctic (Lubinski et al., 2001), or ostracods Bythocythere constricta., Cytheropteron arcuatum, C. champlainum, C. porterae, C. paralatissium, C. tumefactum, Krithe hunti (see Supplementary material). Similar assemblage occurs in unit B1 of core 4-PC1. How to explain the co-existence of these ecologically different species? I would rather assign these assemblages to the environments on a relatively steep slope of the East Siberian Sea or Herald Canyon with paleodepths of 50-60 m, but in close proximity to the paleocoast from where the shallow-water species were either transported downslope with slides or ice-rafted. A "slide event" assemblage was recorded in core PS51/154-11 from the Laptev Sea slope (Taldenkova et al., 2013) at around 15 cal.ka which contained deep-water foraminifers and ostracods along with river-proximal foraminifers, C. macchesneyi and even freshwater ostracod Iliocypris bradii. According to such an assumption, around 12-12.5 cal.ka the sea-level position in both localities was close to -60 m.

- The only "true" shallow-water assemblage dominated by river-proximal species is the one in unit B2 of core 4-PC1, but its age is determined by extrapolation and not supported by any AMS14C dating. In the Laptev Sea, similar fossil assemblages with river-proximal species and Elphidium clavatum among foraminifers, C. macchesneyi, P. pseudopunctillata, H. sorbyana among ostracods and brackishwater molluscs Portlandia aestuariorum and Cyrtodaria kurriana were found in basal sediment units of cores from the outer and middle shelf retrieved from river paleovalleys (Taldenkova et al., 2005, 2008; Stepanova et al., 2012). These assemblages likely dwelled at water depths not exceeding 10 m in former river estuaries during their initial flooding by the transgressing sea. Depending on water depth of these cores that ranges between 60 and 45 m, the ages of these assemblages vary between 12.3 and 10.2 cal.ka. Particularly, in core PS51/159-10 from the Khatanga paleovalley (water depth 60 m) the estuarine assemblage occurs below 400 cm and dates back to 12-12.3 cal.ka. This allows assuming the sea level to be positioned at about -55 m around 12 cal.ka which is consistent with the model estimations and many other lines of evidence from different

Arctic regions including the Hope Valley on the Chukchi Sea (Keigwin et al., 2006).

Some minor corrections and typos: - The title might include not only the East Siberian, but also Chukchi Sea margin, as the Herald Canyon formally belongs to the Chukchi Sea.

- In Fig. 5, the plot of E. incertum percentage should be shown against X-axis range 0-40%, otherwise the visual impression is that H. orbiculare is more abundant than E. incertum, which is not the case.

- In the abstract, 6th sentence from below – the word "during" should be shifted to the right position.

- P. 6, 6th line from top – "East Siberian Sea margin" should be changed to "Chukchi Sea margin". The same correction should be made for Fig. 7 caption.

---

## Author Comment (AC1) · 20 Jun 2017

**Response: See text below**

Based on the interpreted age of 13.5ka at the base of the C1 record, the authors then conclude that regional sea-level was 40-50m lower than geophysical models predict, but that discrepancy is entirely based on the assumption that the age extrapolation to the base of the core is correct. A simpler explanation is that it is not correct, and that the basal sediments are older, a possibility that is not considered.
The authors mention (without discussion) that "Unit B' has 2 facies (B1 and B2). The first time this is mentioned is in Section 4.1. This "lithologic transition" does not enter into their assumption of a linear sedimentation rate below the lowest date (which appears to be in unit B1) but may explain why the sediments towards the base of the core are older than assumed. It also appears that the 417cm sample is immediately below a hiatus of unknown duration in the core. A hiatus in the record seems highly likely. From ~11.5-11.0ka B.P., (MWP-1B) sea level rose by 16m (>3m/century). During this time, water depths at the core site would have been quite shallow, and it is hard to imagine that sediment deposition in this dynamic environment was not completely disturbed. It also seems unlikely that the ~8500 B.P date (on unidentified organics) is correct as this would imply a dramatic reduction in sediment deposition from ~2835 to ~8500 BP., followed by a sharp increase. The discussion of core 20-GC [Section 3.2] is bizarre as there is no consistency in the dates on that record, and the authors simply decide to ignore older ages as being reworked, concluding that the entire record is "probably around 11ka". Similar logic is not applied to old dates on samples dated in cores 23-GC and 24-GCâ˘Tthese are ˇ accepted as correct. There is also a puzzling use of reservoir correctionsâA˘T300 years for the upper section, ˇ but only 50 years for the lower section. In a recent paper–on which the first author here was a co-author (Poirier et al, 2012, Marine Micropaleontology) a reservoir age of 1,000 years was used for samples >10,000 years, as Hanslik et al., 2010 (QSR) also did. One might expect that restricted circulation in the Canada Basin, prior to the opening of Bering Strait, would result in "old water" in this area, requiring a bigger C2 reservoir correction. That would shift the age of the radiocarbon dates in Unit B1 towards younger calibrated ages. Other points: Inconsistencies in core IDs (in text, Figure). Also appear to be errors in Lab ID of C-14 dates in Table [some numbers are duplicated

and Beta-455001 in Fig 4 not listed]. This often makes it a challenge to follow the arguments in the paper. The conclusions reached in the paper are unconvincing. Interactive comment on Clim. Past Discuss., doi:10.5194/cp-2017-19, 2017.

**Response: Before addressing each point individually, let us recount the data presented and interpretations: We interpreted a shallow water sediment sequence 500 to 413 cm core depth with 5 calibrated C14 ages centered on 12 cal ka as being deposited in shelf environments during the Younger Dryas. An abrupt stratigraphic break [hiatus, condensed zone] at 413-400 cm, documented by faunal, geochemical, physical properties and consistent with well mapped geophysical units along an extended region of the Siberia shelf slope break, is dated near the end of the YD ~ 11.2 to 11.0, regardless of what delta R reservoir correction one uses. The lowermost part of the core 600 to 500 cm is not dated.- see new paragraph below discussing this.**
**The upper 400 cm of the core, part of the Holocene, is not germane to the topic of the paper, late glacial sea level, and the new radiocarbon dates are not critically assessed.**

**In light of prior studies of sedimentation deposited during rapid marine transgressions in coastal settings, this record is a textbook example of complex patterns that are not necessarily easy to interpret, but similar to what you find in places like the Black Sea, Chesapeake Bay, the Gulf of Mexico, Sunda Shelf, Tampa Bay and most post glacial Marine settings flooded due to glacio-isostacy. In fact our study region has resemblances to tropical coral reef records of sea level where the evidence for rapid SL is the lack of U-series ages and stratigraphically jumbled coral rubble in key core intervals ! In addition, paleodepth estimates from corals are highly dependent on the particular coral genus studied and in the Pacific can be quite large. So the review seems to be ignoring the broader understanding of the stratigraphy and sedimentation along continental or island margins during rapid transgressions. Nonetheless, we shall try to accommodate his/her concerns.**

**The reviewer is correct. Arctic Ocean sediment records involve dating uncertainty. Whether this problem is unique to the Arctic, to radiocarbon dating, to semi-enclosed basins, or any other marine body of water is arguable. Nonetheless, we agree with the reviewer there is a need to urge caution and we copied below, in smaller font and underlined, our response to the other reviewer who raised similar points. Note however**
   **a) using different delta R values on the date near the main transition, Unit B/A boundary, 400-413 cm core depth, changes the age only slightly (a range of 118 years, from 50, 300, 500 year delta R values) (see Supplementary Figure S1 in Jakobsson et al. CP this volume).**
   **b) the interval 420-500 cm 4-PC1 core depth has 5 dates with calibrated age ranges almost all within the YD from ~10.8 to 12.5 cal ka. The sediments contain shelf faunal assemblages.**

c) we inserted this sentence on page 6: "The core 4-PC1 river-intermediate assemblage centered about 12 ka suggests an early to mid Younger Dryas transgression of this region, although the rate of transgression cannot be quantified from the faunal shifts alone, and additional study of the age of the lowermost sediments in 4-PC1 is needed."

Copied from response to Taldenkova review: The use of a particular reservoir correction in the Arctic has been contentious for years and we do not deny there may be several choices both for calibration [see Hanslik et al. 2013 QSR] and choice of material dated. For our particular Siberian and Chukchi margin cores, we refer to the papers of Pearce et al. and Jakobsson et al., both in this CP volume, for our rationale in using a lower delta R number (50 yrs) for the pre-Holocene/Deglacial than for the Holocene (200 yrs). In our own text, this is made clear on page 4. In Jakobsson et al. Supplement Fig 1, using 3 different delta R values (50, 300, 500 yrs) for NOSAMS date 131218 results in about 118 year range in calibrated ages (11,065, 10,788, 10,547 years) at the time the Bering Sea was flooded, roughly 11,000 years ago. The ages on the dated sections of the SWERUS cores may or may not be equivalent to those from the Laptev Sea.

And...
We noted the age uncertainty in the revision; calcareous fossils were not abundant enough below this level to obtain an AMS data. The text reads "possibly" in regard to marking the onset of the YD.

Response to reviewer 1 regarding age below 500 cm. The reviewer suggests an older age for the 600-500 cm interval, suggesting our parsimonious assumption of extrapolating down core is wrong. But he/she offers no alternative age? 15 ka? 20 ka? (it must be an age that is consistent with the shallow-water nearshore faunas). The reviewer also proposes that reworking of faunas and/or dated material occurred, but on what evidence? But there is excellent consistency in all chemical, physical, microfaunal proxies from this core (see other papers on core 4-PC1 in the CP volume). We note also that another reviewer proposed the opposite of reworking, instead downslope transport as she had observed in the Laptev Sea. But the Siberian margin 4-PC1 core lacks evidence of reworking and downslope movement. In contrast, core 20-GC1, obviously has rapid sedimentation, an inadequate age model, sediment mixing, but nonetheless it recovered 35 cm of sediment containing shelf microfaunas dated at 13.2 to 11 cal ka.

We clarify age uncertainty in the revision by inserting the following paragraph, which hopefully will spur more research:

"Before interpreting the Siberian and Chukchi Sea deglacial sea level chronology, it is useful to examine the broader patterns of LGM and deglacial sedimentation in the Arctic Ocean for context. Marine sediments deposited during the last glacial maximum are uncommon in the central Arctic Ocean

due to the extensive sea ice and ice shelf cover.  For example, Polyak et al. (2004) documented a hiatus between 19 and 13 ka in several cores from the western Arctic. In a compilation of 199 new and published calibrated radiocarbon dates from the central Arctic Ocean, Poirier et al. (2012) found similar results: no dates at 21-22 ka, 4 total from 19-15 ka, 4 dates from 14-15 ka, 5 dates from 13-14 ka, and a spike up to 13 dates between 12-13 ka. Several studies of Arctic Ocean margins have recovered deglacial sediments. Taldenkova et al. (2013) found the earliest deglacial dates of 15.34 and 15.37 ka in core PS51/154-11 at 270 mwd in the Laptev Sea.  These correspond to the first appearance of common benthic foraminifera.  Scott et al. (2009) dated sediments from piston core PC750 (1000 mwd) off the Mackenzie Trough, on the Canadian margin, at 11.3 cal ka at 180 cm and 13.3 ka at 380 cm.  Benthic foraminifera first become common in core PS750 at ~11.3 ka.   In core PS2138-1 (995 mwd) from the Barents Sea slope, north of Spitzbergen, Wollenburg et al. (2004, see also Matthiessen et al. 2001, Norgaard-Pedersen 2003) dated one of the more complete LGM- deglacial sequences with nine calibrated radiocarbon ages from 23.88 to 15.52 ka from 275 to 65 cm core depths. Unlike the Laptev, Siberian and Canadian margins, relatively continuous sedimentation in this region during this period reflects complex changes in productivity and oceanography during Greenland Stadials GS-2 (21-14.6 ka) and GS-1 (14.6-11.6 ka, Bølling-Allerød, Younger Dryas) largely due to changes in inflowing warm Atlantic Water and the West Spitzbergen Current.   In sum, these few examples show that the earliest ages for deglacial sedimentation and preservation of abundant benthic microfaunas (and by inference productive benthic ecosystems) varies along different Arctic Ocean margins. In regards to the undated interval in 4-PC1 core from 609-500 cm core depth, this means that pending further investigations, we cannot completely exclude the possibility that the lowermost sediments below 500 cm core depth in 4-PC1 are older than ~13 ka.”

---

## Author Comment (AC3) · 20 Jun 2017

Thanks for the excellent comments, the response is in the attached file

Please also note the supplement to this comment:
http://www.clim-past-discuss.net/cp-2017-19/cp-2017-19-AC3-supplement.pdf

———————————————

---

## Author Comment (AC4) · 14 Jul 2017

July 14, 2017

Editors, Climate of the Past,

Dear Carlo,

On behalf of my co-authors, we submit a revised manuscript entitled "Deglacial sea-level history of the East Siberian and Chukchi Sea Margins". We already uploaded detailed response to 3 reviewers. In addition, in response to your own comments about the radiocarbon chronology, we made final revisions, shown in the tracked file. You will see the following important changes: 1. Section 4.1 describes how, based on a review of Arctic deglacial records, it is highly unlikely the age of the undated meter

of sediment in 4-PC1 core could be much older than about 13.5 simply because the Arctic Ocean did not experience much sedimentation as ice cover was retreating, at least based on our current knowledge and radiocarbon data. 2. However, we recognize there is uncertainty, and wrote this sentence: "In regards to the undated interval in 4-PC1 core from 609-500 cm core depth, this means that, pending further investigations, we cannot completely exclude the possibility that the lowermost sediments below 500 cm core depth in 4-PC1 are older than ~13.5 ka." 3. We also updated references, including the now published companion Jakobsson et al. paper, which also addresses radiocarbon dating, in this CP volume. 4. We will upload the Appendix data, which includes microfossil species data, with help from Copernicus.

We look forward to your feedback on the new revision.

Sincerely,

Dr. Thomas M. Cronin
* * *
[Figure]

**Fig. 1.** Fig. 1

---

## Author Comment (AC2)

2017 Interactive comment on Clim. Past Discuss., doi:10.5194/cp-2017-19, 2017.

The paper "Deglacial sea-level history of the East Siberian Sea Margin" has the aim
to reconstruct and estimate, from sediment cores, the paleo-depth in the eastern
margin using benthic foraminifera and ostracode species. Thought the results are
discussed properly and they could be relevant for a large audience, I am not
convinced by the dating of the cores. Without a confident age calibration all the
discussion given in the manuscript are of secondary importance. The results
obtained by the authors can be much improved if the age scale and the confidence
associate with the dating of the core will be improved. The chronology of the lower
section of core PC1 is based only on 5 points (6 with the outlier) where four of them
suggest a similar age, though they cover 33 cm of the core. Considering this
uncertainties and lack of age calibration points in the bottom C1 part of the core I
would be careful in assume an almost linear relationship depth vs age to date the
bottom part, especially for the time periods investigate.

Response: We agree it is important to discuss the uncertainty about a) C14 dated
intervals and reservoir corrections and b) the lower part of 4-PC1 core, 500-600 cm
core depth which is not directly dated. Here are revisions to address these.
a) **The use of a particular reservoir correction in the Arctic has been
contentious for years and we do not deny there may be several choices both
for calibration [see Hanslik et al. 2013 QSR] and choice of material dated.  For
our particular Siberian and Chukchi margin cores, we refer to the papers of
Pearce et al. and Jakobsson et al., both in this CP volume, for our rationale in
using a lower delta R number (50 yrs) for the pre-Holocene/Deglacial than for
the Holocene (200 yrs).  In our own text, this is made clear on page 4. In
Jakobsson et al. Supplement Fig 1, using 3 different delta R values (50, 300,
500 yrs) for NOSAMS date 131218 results in about 118 year range in
calibrated ages (11,065, 10,788, 10,547 years) at the time the Bering Sea was
flooded, roughly 11,000 years ago.  The ages on the dated sections of the
SWERUS cores may or may not be equivalent to those from the Laptev Sea.**

b) **We noted the age uncertainty in the revision; calcareous fossils were not abundant
enough below this level to obtain an AMS data. The text reads "possibly" in regard to
marking the onset of the YD.**
**Also, we added a section of text to this effect.**
**"Marine sediments deposited during the last glacial maximum are uncommon
in the central Arctic Ocean due to the extensive sea ice and ice shelf cover.  For
example, Polyak et al. (2004) documented a hiatus between 19 and 13 ka in
several cores from the western Arctic. In a compilation of 199 new and
published calibrated radiocarbon dates from the central Arctic Ocean, Poirier
et al. (2012) found similar results: no dates at 21-22 ka, 4 total from 19-15 ka,
4 dates from 14-15 ka, 5 dates from 13-14 ka, and a spike up to 13 dates**

**between 12-13 ka. Several studies of Arctic Ocean margins have recovered deglacial sediments. Taldenkova et al. (2013) found the earliest deglacial dates of 15.34 and 15.37 ka in core PS51/154-11 at 270 mwd in the Laptev Sea. These correspond to the first appearance of common benthic foraminifera. Scott et al. (2009) dated sediments from piston core PC750 (1000 mwd) off the Mackenzie Trough, on the Canadian margin, at 11.3 cal ka at 180 cm and 13.3 ka at 380 cm. Benthic foraminifera first become common in core PS750 at ~11.3 ka. In core PS2138-1 (995 mwd) from the Barents Sea slope, north of Spitzbergen, Wollenburg et al. (2004, see also Matthiessen et al. 2001, Norgaard-Pedersen 2003) dated one of the more complete LGM-deglacial sequences with nine calibrated radiocarbon ages from 23.88 to 15.52 ka from 275 to 65 cm core depths. Unlike the Laptev Sea, Siberian and Canadian margins, relatively continuous sedimentation in this region during this period reflects complex changes in productivity and oceanography during Greenland Stadials GS-2 (21-14.6 ka) and GS-1 (14.6-11.6 ka, Bølling-Allerød, Younger Dryas) largely due to changes in inflowing warm Atlantic Water and the West Spitzbergen Current. In sum, these few examples show that the earliest ages for deglacial sedimentation and preservation of abundant benthic microfaunas (and by inference productive benthic ecosystems) varies along different Arctic Ocean margins. Thus, pending further investigations, we cannot completely exclude the possibility that the lowermost sediments below 500 cm core depth in 4-PC1 are older than ~13 ka."**

The periods between 10 ky and 15 ky BP in the Arctic, have been characterize by rapid climate changes brought to different climate condition. In this period we can distinguish three different climate phases, the Younger Dryas, the Bølling-Allerød interstadial and the early Holocene. These periods could have been characterized by different deposition and sedimentation rate making the assumption of a linear sedimentation rate below the lowest date not convincing. The core 20-GC1 shows a similar age between 55 to 90 cm depths. Could the continuity of these sediments be disturbed and reshuffle by the Fennoscandinavia ice sheet movement?
**Response: Yes reworking of dated material in 20-GC1 is possible and we clarify this point, although we don't know if ice sheet or ice shelf margins are involved, or sedimentary or biological processes on the upper slope.**

Without additional information improving the core chronology, especially for the bottom part of the PC1 core, the discussion present by the authors are not convincing.
**Response. As stated above, our focus was on the significance of the well-documented [multiproxy, geophysics etc] transition in 4-PC1 core at 413-400 cm core depth. The more subtle microfaunal shift in the interval 500-413 cm is well-dated by 5 dates and the delta R correction is defended here and in other papers in this volume. However, the reviewer feels it is necessary to also date the oldest deglacial sediments at 600-500 cm core depth, which unfortunately lack enough suitable material. The new paragraph added to the**

text (see above) concedes this. In light of reviewed records from other Arctic margins providing previously published ages of deglacial sediments, the upshot is that rarely are sediments dated from the LGM up to 14-15 kyr from the Arctic except near regions north of Spitzbergen. This is most likely due to extensive thick ice shelf and sea ice cover until well after the Bølling-Allerød.

We added the following sentence at the end of the discussion:

"Arctic Ocean deglaical sea level history remains incomplete, however, and it will be necessary in future studies to extend the deglacial sea level record back to the early stages of deglaciation prior to the Younger Dryas. "

In sum, with all due respect, Arctic sedimentary chronology and depositional history cannot simply be viewed from the standpoint of radiocarbon dating. It must be interpreted in light of many factors: LGM ice extent, geophysical profiles of cored locations, physical stratigraphy, ice rafting, the rate of sea level rise, sedimentation rates, spatial variability in Arctic sea ice, land ice, ice shelves, paleoceanography, core location with respect to inflowing Atlantic Water (and Pacific water through the Bering Strait), riverine freshwater and sediment input, and submarine geomorphology of the continental margin. The context for the Siberian/Chukchi margin sea-level record in this paper is given in the transects of cores shown in Figure 2 and 3 and in additional papers on these cores.